# Challenges of Real-World Reinforcement Learning

**Gabriel Dulac-Arnold** [1]   **Daniel Mankowitz** [2]   **Todd Hester** [2]

## Abstract

Reinforcement learning (RL) has proven its worth in a series of artificial domains, and is beginning to show some successes in real-world scenarios. However, much of the research advances in RL are often hard to leverage in real-world systems due to a series of assumptions that are rarely satisfied in practice. We present a set of nine unique challenges that must be addressed to productionize RL to real world problems. For each of these challenges, we specify the exact meaning of the challenge, present some approaches from the literature, and specify some metrics for evaluating that challenge. An approach that addresses all nine challenges would be applicable to a large number of real world problems. We also present an example domain that has been modified to present these challenges as a testbed for practical RL research.

## 1. Introduction

Reinforcement learning (RL) (Sutton & Barto, 2018) is a powerful algorithmic paradigm encompassing a wide array of contemporary algorithmic approaches (Mnih et al., 2015; Silver et al., 2016; Hafner et al., 2018). RL methods have been shown to be effective on a large set of simulated environments (Mnih et al., 2015; Silver et al., 2016; Lillicrap et al., 2015; OpenAI), but uptake in real-world problems has been much slower. We posit that this is due in large part to a too-wide divide between the casting of current experimental RL setups and the generally poorly defined realities of real-world systems.

In this paper, we present the main challenges that make RL in the real-world more difficult than RL in research. At a high-level these challenges are:

1. Training off-line from the fixed logs of an external behavior policy.
2. Learning on the real system from limited samples.
3. High-dimensional continuous state and action spaces.
4. Safety constraints that should never or at least rarely be violated.
5. Tasks that may be partially observable, alternatively viewed as non-stationary or stochastic.
6. Reward functions that are unspecified, multi-objective, or risk-sensitive.
7. System operators who desire explainable policies and actions.
8. Inference that must happen in real-time at the control frequency of the system.
9. Large and/or unknown delays in the system actuators, sensors, or rewards.

While there has been research focusing on these challenges individually, there has been little research on algorithms that address all of these challenges together. We hope that these challenges can guide researchers towards developing more applicable RL algorithms. For each challenge above, we motivate its importance, specify it, present approaches for addressing the challenge, and provide methods for evaluating that particular challenge. Finally, we illustrate these challenges on a toy-problem, a control suite (Tassa et al., 2018) task modified to present all of the challenges described above.

We consider control systems grounded in the physical world, optimization of software systems, and systems that interact with users such as recommender systems and smart phones. These systems can range in size from a small drone to a datacenter, in complexity from a one-dimensional thermostat to a self-driving car, and in cost from a calculator to a spaceship. In all these scenarios there are recurring themes: there is rarely a good simulator, the systems are stochastic & non-stationary, have strong safety constraints, and running them is expensive and/or slow. This is very different from training on a simulated environment where data is effectively unlimited, consequences for poor actions are non-existent and system dynamics are clean and often deterministic.

Although current RL algorithms can learn superhuman policies for systems we can properly simulate (Silver et al., 2016; Mnih et al., 2015; OpenAI), for many real-world sys-

---

[1]Google Research, Brain Team [2]DeepMind. Correspondence to: Gabriel Dulac-Arnold <dulacarnold@google.com>.

*Reinforcement Learning for Real Life (RL4RealLife) Workshop in the 36th International Conference on Machine Learning*, Long Beach, California, USA, 2019. Copyright 2019 by the author(s).

tems, not only is there no existing simulator, but building one can be extremely difficult. Many interesting systems are either too complex to model properly (datacenter cooling plants or deformable object manipulation tasks), or sufficiently varied (arbitrary object assembly with the same robot arm) that modelling each instance would be impractical. This lack of available simulators means learning must be done using data from the real system, and all acting and exploring must be done on the real system. Thus, we cannot simply collect massive datasets to solve these challenges, nor can we ignore safety during training.

To deploy RL to a real production system, robust evaluation is required. Many research papers in RL look at average episodic return to evaluate the quality of their agent (although they train with discounted return). This makes sense when the only optimization criteria is the return itself, however in real systems there are other aspects of agent behavior which are equivalently important. In many cases, it may be important to evaluate performance for the worst case user, or the worst case object for manipulation, rather than the average reward. For many real world applications, respecting safety can be much more important than maximizing returns. Additionally, because the global reward function is generally a balance of multiple sub-goals (*e.g.* reducing both time-to-target and energy use), a proper evaluation should explicitly separate the individual components of the reward function to better understand the policy's tradeoffs.

## 2. Practical Challenges

In this section we present a series of practical challenges that appear when using RL on real world systems. Not all of these challenges will be present in every real system, but in many cases all of the challenges are present to some degree. To guide practitioners working on applications of RL, we present current research directions from the literature for each challenge. To guide researchers who wish to research only a subset of challenges at a time, we present evaluation criteria for that particular challenge. We believe that an RL algorithm that addresses all of these challenges would be applicable to a vast number of real world problems.

For this work, we assume the standard Markov Decision Process (MDP) formulation. An MDP is represented by a tuple $\langle \mathcal{S}, \mathcal{A}, P, r, \gamma \rangle$ where $\mathcal{S}$ is the state space, $\mathcal{A}$ the action space, $P$ is the stochastic transition function $p(s' \mid s, a)$, and $r(s, a)$ is the reward function which returns a reward for given state-action. A tuple of experience is of the form $(s_t, a_t, r_t, s_{t+1})$. The policy $\pi$ produces an action as $a_t \sim \pi(\cdot|s_t)$. Our notation describes a discrete state and action space but generalizes to continuous ones. Later, we consider modifications of this formalism for safety constraints, robustness, non-stationarity and partial observability. As there will likely be multiple iterations of the policy through time we index them as $\pi_i$ to indicate the learning it-

eration. An existing system is often controlled by an existing policy, either in the form of human operators or black-box controllers; we call this a *behavior* policy and denote it $\pi_B$.

### 2.1. Batch Off-line and Off-Policy Training

As mentioned above, many systems cannot be trained on directly and need to be learned from fixed logs of the system's behavior. In many cases, we are deploying an RL approach to replace a previous control system, and logs from that policy are available. In future training iterations, batches of data will be available from the most recent iteration of the control algorithm. This setup is an off-line and off-policy training regime where the policy needs to be trained from batches of data. We begin by proposing a basic framework and then discuss possible design choices.

We consider an ordered set of experience tuples produced by policy $\pi_B$, $\mathcal{D}_{\pi_B} = [\{s_0, a_0, r_0, \ldots, s_T, a_T, r_T\}_{1 \leq i \leq n}]$. The initial policy $\pi_0$ is trained from data from the previous controller of the system, $D_{\pi_B}$. $\pi_0$ generates data for $D_{\pi_0}$ which is used to train $\pi_1$ and so-on [1]. The details of policy training are left up to the implementer. The general framework is that of batched reinforcement learning (Scherrer et al., 2012), which we recall in Algorithm 1 for clarity. We note that the RL algorithm used to train the policy after each batch is not restricted; it could be policy iteration, value function based, policy gradient, etc. (Sutton & Barto, 2018).

---

**Algorithm 1** Batch RL Training

1: **procedure** BRT($\mathcal{D}_{\pi_B}$, Train, $N$, $L$)
2: $\quad \pi_0 \leftarrow$ Train($D_{\pi_B}$)
3: $\quad$ **for** $i = 0, \cdots, N$ **do**
4: $\quad\quad \mathcal{D}_{\pi_i} \leftarrow \{\}$
5: $\quad\quad$ **for** $t = 0, \cdots, L$ **do**
6: $\quad\quad\quad a_t \leftarrow \pi_i(s_t)$
7: $\quad\quad\quad \mathcal{D}_{\pi_i} \leftarrow \mathcal{D}_{\pi_i} \cup (s_t, a_t, r_t(s_t, a_t))$
8: $\quad\quad\quad s_{t+1} \sim P(\cdot \mid s_t, a_t)$
9: $\quad\quad$ **end for**
10: $\quad\quad \pi_{i+1} \leftarrow$ Train($\mathcal{D}_{\pi_i}$)
11: $\quad$ **end for**
12: **end procedure**

---

For a production system where drops in performance could be very costly, we want to ensure that the new policy improves upon the previous policy. Estimating the policy's performance without running it on the real system is termed off-policy evaluation (Precup et al., 2000). Off-policy evaluation becomes more challenging as the difference between the policies and the resulting state distributions grows.

The simplest approach to off-policy evaluation is importance sampling (Precup et al., 2000), which accounts for the difference between the behavior and target policies. Alternatively, the direct method learns a transition model and

---

[1] We consider that any mix of previous experience is acceptable for training the policy, one does not have to limit training to $D_{\pi_{t-1}}$.

uses that for evaluation. Doubly-robust estimators (Dudík et al., 2011; Jiang & Li, 2015) combine both and get the best evaluations from both worlds. There are many more approaches such as MAGIC (Thomas & Brunskill, 2016) or more robust doubly robust (Farajtabar et al., 2018) that can be considered as well.

The performance of the initial policy $\pi_0$ often dictates whether access to a system will be granted by system owners as there is usually a minimum performance threshold the system must respect. Therefore, an important quantity to evaluate for a new learning algorithm is the warm-start performance given the behavior policy's data:

$$J^{start} = R(\text{Train}(D_{\pi_B})), \quad (1)$$

where R is the cumulative return from the policy $\text{Train}(D_{\pi_B})$. Evaluating the training algorithm's performance on different sizes of $\mathcal{D}_{\pi_B}$, as well as for different quality behavior policies can help understand the algorithm's ability at finding a starting policy even in the face of sub-optimal, over-fit, or insufficient data.

## 2.2. Learning On the Real System from Limited Samples

Unlike much of the research performed in deep reinforcement learning (Mnih et al., 2015; Hester et al., 2018), real systems do not have separate training and evaluation environments. All training data comes from the real system, and the agent cannot have a separate exploration policy during training as its exploratory actions do not come for free. Instead, the agent must perform reasonably well and act safely throughout learning. For many systems, this means that exploration must be limited, and the resulting data is low-variance – very little of the state space may be covered in the logs. In addition, since there is often only one instance of the system, approaches that instantiate hundreds or thousands of environments to collect more data for distributed training are usually not compatible with this setup (Horgan et al., 2018; Espeholt et al., 2018; Adamski et al., 2018).

Almost all of these real-world systems are either slow-moving, fragile, or expensive enough that the data they produce is costly, and policy learning must be data-efficient. In the case where there are off-line logs of the system, these might not contain anywhere near the amount of data or data coverage that current RL algorithms expect. Learning iterations on a real system can take a long time, as slower control frequencies might range from 1-hour to multi-month timesteps, and reward horizons could be on the order of months (e.g. online advertisement, drug therapies). Even in the case of higher-frequency control tasks, the learning algorithm needs to learn quickly from potential mistakes without needing to repeat them multiple times before fixing them. Thus, learning on a real system requires an algorithm to be both sample-efficient and performant.

There are a number of related works that deal with RL on real systems and, in particular, focus on sample efficiency. One such work is Model Agnostic Meta-Learning (MAML) (Finn et al., 2017) that focuses on learning about tasks within a distribution and then, with few shot learning, quickly adapting to solving a new in-distribution task that it has not seen previously. Bootstrap DQN (Osband et al., 2016) learns an ensemble of Q-networks and uses Thompson Sampling to drive exploration and improve sample efficiency, and PILCO (Deisenroth & Rasmussen, 2011) uses Gaussian processes to efficiently model a system and train a policy.

Another approach to improving sample efficiency is to use expert demonstrations to bootstrap the agent, rather than learning from scratch. This approach has been combined with DQN (Mnih et al., 2015) and demonstrated on Atari (Hester et al., 2018), as well as combined with DDPG (Lillicrap et al., 2015) for insertion tasks on robots (Vecerík et al.). Recent Model-based deep RL approaches (Hafner et al., 2018; Chua et al., 2018), where the algorithm plans against a learned transition model of the environment, show a lot of promise for improving sample efficiency. A common approach is to learn ensembles of transition models and use various sampling strategies from those models to drive exploration and improve sample efficiency (Hester & Stone, 2013; Chua et al., 2018; Buckman et al., 2018).

To evaluate the data efficiency of a particular method, a simple yet useful measure is to look at the amount of data necessary to achieve a certain performance threshold:

$$J^{eff.} = \min |\mathcal{D}_i| \text{ s.t. } R(\text{Train}(D_i)) > R_{\min}, \quad (2)$$

where $R_{\min}$ is the desired performance threshold. Similar approaches have been used in model-based RL literature to demonstrate efficiency (Hafner et al., 2018; Chua et al., 2018).

## 2.3. High-Dimensional Continuous State and Action Spaces

Many practical real world problems have large and continuous state and action spaces. For example, consider the huge action spaces in recommender systems (Covington et al., 2016), or the number of sensors and actuators to control cooling in a Google data center (Evans & Gao; Evans et al., 2018). These large state and action spaces can present serious issues for traditional RL algorithms, as identified in (Dulac-Arnold et al., 2015).

There are a number of recent works focused on addressing this challenge. Dulac-Arnold et al. present an approach based on generating a vector for a candidate action and then doing nearest neighbor search to find the closest real action available. Zahavy et al. propose an Action Elimination Deep Q Network (AE-DQN) that uses a contextual

bandit to eliminate irrelevant actions. He et al. present the Deep Reinforcement Relevance Network (DRRN) for evaluating continuous action spaces in text-based games. Finally, Chandak et al. proposes a method to learn action embedding according to their effects on state transitions.

Evaluating a policy in the face of large action spaces should consider both the number of actions and the quality of the metric over the action space: well-organized action spaces are easier to reason with than smaller but poorly ordered spaces. Even if embeddings are learned, the inherent relationship between actions should be varied when evaluating.

## 2.4. Satisfying Safety Constraints

Almost all physical systems can destroy or degrade themselves and their environment. As such, considering these systems' safety is fundamentally necessary to controlling them. Safety is important during system operation, but also during exploratory learning phases as well. These could be safety considerations either of the system itself (limiting system temperatures, contact forces or maintaining minimum battery levels) or of its environment (avoiding dynamic obstacles, limiting end effector velocities). There may exist a fallback watchdog controller, which would take over if the learned policy violates the safety constraints, but we consider that it should not be explicitly relied upon.

Recent work in RL safety (Dalal et al., 2018; Achiam et al., 2017) has cast safety in the context of Constrained MDPs (CMDPs) (Altman, 1999), and we will concentrate on pre-defined constraints on the environment in this context. Constrained MDPs define a constrained optimization problem and can be expressed as: $\max_{\pi \in \Pi} R(\pi)$ subject to $C^k(\pi) \leq V_k, k = 1, \ldots, K$.

Here, $R$ is the cumulative reward of a policy $\pi$ for a given MDP, and $C^k(\pi)$ describes the incurred cumulative cost of a certain policy $\pi$ relative to constraint $k$. The CMDP framework describes multiple ways to consider cumulative cost of a policy $\pi$: the total cost until task completion, the discounted cost, or the average cost. Specific constraints are defined as $c_k(s, a)$.

The CMDP setup allows for arbitrary constraints on state and action to be expressed. In the context of a physical system these can be as simple as box constraints on a specific state variable, or more complex such as dynamic collision avoidance constraints. One major challenge with addressing these safety concerns in real systems is that safety violations will likely be very rare in logs of the system. In many cases, safety constraints are assumed and are not even specified by the system operator or product manager.

An alternative to CMDPs is budgeted MDPs (Boutilier & Lu, 2016; Carrara et al., 2018). While for a CMDP, the constraint level $V_k$ is given, for budgeted MDPs, it is unknown. Instead, the policy is learned as a function of constraint level.

Then the user can examine the trade-offs between expected return and constraint level and choose the constraint level that best works for the data. This more closely matches the common real-world scenario where the constraints may not be absolute, but small violations may be allowed for a large improvement in expected returns.

Recently, there has a been a lot of work focused on the problem of safety in reinforcement learning. One focus has been the addition of a safety layer to the network (Dalal et al., 2018; Pham et al., 2017). This type of approach has enabled an agent to learn a task with zero safety violations as well as transfer some problems to the real world. These approaches focus on safety during training. There are other approaches (Achiam et al., 2017; Tessler et al., 2018; Bohez et al., 2019) that learn a policy that violates constraints during training but produce a *trained* policy that respects the safety constraints. Thomas (2015) presents the notion of 'safe RL' where the algorithm searches for a new, improved policy and ensures that the probability of a 'bad policy' being proposed is low. In addition, this work computes a high confidence lower bound on the performance of an evaluation policy.

Other RL approaches include using Lyapunov functions to learn safe policies (Chow et al., 2018) and safe exploration strategies that predict the safety of neighboring states (Turchetta et al., 2016; Wachi et al., 2018). A Probabilistic Goal MDP (Mankowitz et al., 2016) is another type of objective that encourages an agent to consider whether it should do something risky for large reward or be more conservative for smaller rewards within a pre-defined period of time.

To evaluate the safety of an RL algorithm, we consider counting the number of safety violations for each individual constraint. Accumulating all these violations into a single number has been proposed previously (Dalal et al., 2018) and provides a good global summary. We propose also maintaining each individual constraint's count of violations:

$$\boldsymbol{J}^{safety}(\pi) = \left( \sum_{i=1}^{T} c_j(s_i, a_i) \right)_{1 \leq j \leq K} \in \mathbb{R}^K, \quad (3)$$

where $K$ is the number of safety constraints in the CMDP. Visualizing this vector's evolution during both training and once running on the environment is essential to understanding the policy's behavior relative to the provided safety constraints.

## 2.5. Partial Observability and Non-Stationarity

Almost all real systems where we would want to deploy reinforcement learning are partially observable. For example, on a physical system, we likely do not have observations of the wear and tear on motors or joints, or the amount of buildup in pipes or vents. On systems that interact with

users such as recommender systems, we have no observations of the mental state of the users. Often times, these partial observabilities appear as non-stationarity (e.g. as a pump's efficiency degrades) or as stochasticity (e.g. as each robot being operated behaves differently). Partially observable problems are typically formulated as a partially observable Markov Decision Process (POMDP). The key difference from the MDP formulation is that the agent's observation $x \in X$ is now separate from the state, with an observation function $O(x \mid s)$ giving the probability of observing $x$ given the environment state $s$.

There are two common approaches to handling partial observability in the literature. First is to incorporate history into the observation of the agent. DQN (Mnih et al., 2015) stacks four Atari frames together as the agent's observation to account for partial observability. An alternate approach is to use recurrent networks within the agent, enabling them to track and recover hidden state. Hausknecht & Stone apply such an approach to DQN, and show that the recurrent version can perform equally well in Atari games when only given a single frame as input. Nagabandi et al. propose an approach modeling the system as non-stationary with a time-varying reward function, and use meta-learning to find policies that will adapt to this non-stationarity.

Much of the recent work on transferring learned policies from simulation to real system also focuses on this area, as the underlying differences between the systems are not observable (Andrychowicz et al., 2018; Peng et al., 2018). Real world systems are often stochastic and noisy compared to most simulated environments. In addition, sensor and action noise as well as action delays add to the perturbations an agent may experience in the real-world setting. There are a number of RL approaches that have been utilized to ensure that an agent is robust to different subsets of these factors. We will focus on the Robust MDP formalism, domain randomization and system identification as frameworks for dealing with noisy, non-stationary systems.

A Robust MDP is defined by a tuple $\langle \mathcal{S}, \mathcal{A}, \mathcal{P}, r, \gamma \rangle$ where $S, A, r$ and $\gamma$ are as previously defined; $\mathcal{P}$ is a set of transition matrices referred to as the uncertainty set (Iyengar, 2005). The objective that we optimize is the worst case value function defined as:

$$J^{worst}(\pi) = \inf_{p \in \mathcal{P}} \mathbb{E}^p \left[ \sum_{t=0}^{\infty} \gamma^t r_t | \mathcal{P}, \pi \right].$$

At each step, nature chooses a transition function that the agent transitions with so as to minimize the long term value. The agent learns a policy that maximizes this worst case value function. Recently, a number of works have surfaced that have shown this formulation to yield robust policies that are agnostic to a range of perturbations in the environment (Tamar et al., 2014; Mankowitz et al., 2018; Shashua & Mannor, 2017). The solutions do tend to be overly conserva-

tive but some work has been done to yield less conservative, 'soft-robust' solutions (Derman et al., 2018).

In addition to the robust MDP formalism, the practitioner may be interested in both robustness due to domain randomization and system identification. Domain randomization (Peng et al., 2018) involves explicitly training an agent on various perturbations of the environment and averaging these learning errors together during training. System identification involves training a policy that can determine online the environment in which it is operating and modify the policy accordingly (Finn et al., 2017; Nagabandi et al., 2018).

We can train a policy to be robust, but the question arises as to how we can evaluate the performance of such a policy. Robustness to noisy measurements (possibly both sensor and action noise), as well as action delays, can be evaluated by executing the policy in $K$ test environments that exhibits these types of perturbations. Comparing the average test performance of the policy across the $K$ perturbed test environments can provide a notion of robustness. That is,

$$J^{robust}(\pi) = \frac{1}{K} \sum_{p \in \mathbf{P}} \mathbf{E}^p \left[ \sum_{i=1}^{T} r(s_i, a_i) \right], \qquad (4)$$

where $p$ is a perturbed test environment within the test set $\mathbf{P}$. This has previously been applied in a number of different works (Mankowitz et al., 2018; Di Castro et al., 2012; Derman et al., 2018). Evaluating the performance of a policy on a stream of constantly changing environment perturbations and its ability to adapt online to these perturbations provides another notion of robustness. In each of these cases, for a given task, experiment designers must decide which dimensions of the perturbations are pertinent and look at the effects of variations in these dimensions on policy performance.

### 2.6. Unspecified and Multi-Objective Reward Functions

Reinforcement learning frames policy learning through the lens of optimizing a global reward function, yet most systems have multi-dimensional costs to be minimized. In many cases, system or product owners do not have a clear picture of what they want to optimize. When an agent is trained to optimize one metric, other metrics are discovered that also need to be maintained or improved. Thus, a lot of the work on deploying RL to real systems is in formulating the reward function, which may be multi-dimensional. Because the global reward function is generally a balance of multiple sub-goals (*e.g.* reducing both time-to-target and energy use), a proper evaluation should explicitly separate the individual components of the reward function to better understand the policy's tradeoffs.

In addition, it may be desired that the policy performs well

for all task instances and not just in expectation. For example, an algorithm deployed to a factory needs to work on every robot, not just on average, and a recommender system must work for every user, not just on average. Therefore, policy quality cannot be summarized by a single scalar describing cumulative reward, but must consider both multiple dimensions of the policy's behavior and the full distribution of behaviors both during training and testing. A typical approach to evaluate the full distribution of reward across groups is to use a Conditional Value at Risk (CVaR) objective (Tamar et al., 2015b), which looks at a given percentile of the reward distribution, rather than expected reward. Tamar et al. show that by optimizing reward percentiles, the agent is able to improve upon its worst-case performance. Distributional DQN (Dabney et al., 2018; Bellemare et al., 2017) explicitly models the distribution over returns, and it would be straight-forward to extend it to use a CVaR objective.

There are a number of works devoted to recovering an underlying reward function from demonstrations, such as inverse reinforcement learning (Russell, 1998; Ng et al., 2000; Abbeel & Ng, 2004; Ross et al., 2011). Hadfield-Menell et al. examine how to infer the truly intended reward function from the given reward function and training MDPs, to ensure that the agent performs as intended in new scenarios. For problems with multi-objective reward functions, there are approaches to learning the pareto-optimal reward function (Roijers et al., 2013), but none of these have been scaled to the deep reinforcement learning setting yet. Van Seijen et al. present an approach that takes advantage of multi-objective reward signals to learn super-human performance in the Atari game Mc-PacMan.

We propose a simple multi-objective analysis of return. If we consider that the global reward function is defined as a linear combination of sub-rewards, $r(s,a) = \sum_{j=1}^{K} \alpha_j r_j(s,a)$, then we can consider the vector of per-component rewards for evaluation:

$$\boldsymbol{J}^{multi}(\pi) = \left( \sum_{i=1}^{T_n} r_j(s_i, a_i) \right)_{1 \le j \le K} \in \mathbb{R}^K. \quad (5)$$

When dealing with multi-objective reward functions, it is important to track the different objectives individually when evaluating a policy. This way, problem stakeholders can understand the different tradeoffs the policy is making and choose which compromises they consider best.

To evaluate the performance of the algorithm across the full distribution of scenarios (e.g. users, tasks, robots, objects, etc.), we independently analyze the performance of the algorithm on each cohort. This is also important for ensuring fairness of an algorithm when interacting with populations of users. Another approach is to analyze the CVaR return rather than expected returns (Tamar et al., 2015b). One evaluation procedure is to determine whether rare catastrophic rewards are minimized (Tamar et al., 2015b;a). Another evaluation procedure is to observe behavioural changes when an agent needs to be risk-averse or risk-seeking such as in football (Mankowitz et al., 2016).

## 2.7. Explainability

Another essential aspect of real systems is that they are owned and operated by humans, who need to be reassured about the controllers' intentions and require insights regarding failure cases. For this reason, policy explainability is important for real-world policies. Especially in cases where the policy might find an alternative and unexpected approach to controlling a system, understanding the longer-term intent of the policy is important for obtaining stakeholder buy-in. In the event of policy errors, being able to understand the error's origins *a posteriori* is essential.

Verma et al. define their policies using a domain-specific programming language, and then use a local search algorithm to distill a learned neural network policy into an explicit program. Additionally, the domain-specific language is verifiable, which allows the learned policies to be verifiably correct. There are many methods to attempt to elicit the intent of deep neural networks (Montavon et al., 2018) which could also be used to understand a learned policy. Additionally, model-based methods with explicit rollouts used for planning (Hafner et al., 2018; Chua et al., 2018) can provide insights on what the policy's 'intent' may have been. In the case of (Verma et al., 2018), evaluation is centered on looking at the performance of the programatically defined policy, since the policy is inherently defined as a human-readable program. Evaluating actual explainability comes down to evaluating how well a human understands the intent of the policy's expression, which can be done through A/B experiments with users on mechanical turk (Poursabzi-Sangdeh et al., 2018).

## 2.8. Real-Time Inference

To deploy RL to a production system, policy inference must be done in real-time at the control frequency of the system. This may be on the order of milliseconds for a recommender system (Covington et al., 2016) responding to a user request or the control of a physical robot, and up to the order of minutes for building control systems (Evans & Gao). This constraint both limits us from running the task faster than real-time to generate massive amounts of data quickly (Silver et al., 2016; Espeholt et al., 2018) and limits us from running slower than real-time to perform more computationally expensive approaches (e.g. some forms of model-based planning).

There has been literature focused on this problem in the case of robotics. Hester et al. present a real-time architecture to do model-based RL on a physical robot that uses Monte

Carlo Tree Search and returns actions when required by the task, but will improve given more time and more rollouts. Other rollout-based approaches like AlphaGo (Silver et al., 2016) will improve with more rollouts, but are not engineered to run at a specific frequency. Wawrzyński analyzes the potential gains to be had by allowing more computation time than is possible on a true real-time system.

### 2.9. System Delays

Finally, most real systems have delays in either the sensation of the state, the actuators, or the reward feedback. For example, Hester & Stone focus on controlling a robot vehicle with significant delays in the control of the braking system. They incorporate recent history into the state of the agent so that the learning algorithm can learn the delay effects itself. Mann et al. look at delays in recommender systems, where the true reward is based on the user's interaction with the recommended item, which may take weeks to determine. They present a factored learning approach that is able to take advantage of intermediate reward signals to improve learning in these delayed tasks.

Hung et al. introduce a method to better assign rewards that arrive significantly after a causative event. They use a memory-based agent, and leverage the memory retrieval system to properly allocate credit to distant past events that are useful in predicting the value function in the current timestep. They show that this mechanism is able to solve previously unsolvable delayed reward tasks. Arjona-Medina et al. introduce the RUDDER algorithm, which uses a backwards-view of a task to generate a return-equivalent MDP where the delayed rewards are re-distributed more evenly throughout time. This return-equivalent MDP is easier to learn and is guaranteed to have the same optimal policy as the original MDP. They improvements using this approach in Atari tasks with long delays.

Evaluation in this context depends on the source of delay. Delays in the state & action domain can be patched onto existing environments, and the correlation between delay magnitude and the policy's cumulative reward can be used to evaluate the policy's robustness to these perturbations.

### 3. Example Environment

In this section, we present an example environment from the DeepMind control suite (Tassa et al., 2018), and the modifications required to make it present all of the challenges for real world RL that we have presented in this paper[2]. We also describe how to perform specific evaluations for each challenge where it makes sense. Our goal is both that this environment and its modifications drive research in real-world RL as well as help evaluate candidate algorithms' applicability to real-world problems.

---

[2]A more complete task description, as well as three other proposed tasks are presented in the Appendix.

**Humanoid** Variables: $\theta$, $u$, $F$

| Type | Constraint |
|---|---|
| Static | Limit joint angles. Enforce uprightness. Avoid dynamic obstacles. |
| Kinematic | Limit joint velocities. |
| Dynamic | Limit foot contact forces. Encourage falls on pelvis. |

*Table 1.* Safety constraints for the Humanoid environment.

We start from the control suite's humanoid task, as it already has a high-dimensional continuous state and action space. Next, we present a set of modifications to the task to make it present all nine challenges.

First, the training must be done in the batch RL regime as presented in Sec. 1. To imitate the situation where we are taking over control from a system with an existing controller, we recommend first training a policy on the task with a different algorithm than the one being evaluated, and then generating a dataset from this policy to serve as the initial $D(\pi_b)$. As the size, quality, and coverage of this dataset can be critical, we suggest evaluating algorithms with a variety of datasets of different sizes, converged performances, and policy entropies.

Second, we are constrained to act on-line in the domain, meaning every training run needs to be considered as if it is on the actual system, thus incurring safety and efficiency penalties. For the third challenge, the humanoid domain already presents a high-dimensional continuous state and action space. Table 1 proposes safety constraints for the humanoid task. In the case of physical systems, we can generally consider that safety constraints involve one or many of static, kinematic and dynamic aspects of the system.

One can modify the simulation to have non-stationarity and partial observability by sampling a new set of domain parameters (frictions, CoGs, masses) from a distribution every episode. This can either be done during both training and evaluation, or only during evaluation depending on the envisioned scenario. For multi-objective reward, we consider rewards for distance moved as well as for energy used in movement. We also consider worse-case reward for each sampled episode; we want the agent to perform well in every perturbation, not just on average.

For explainability, there is not a modification to the domain, but evaluation must be performed qualititatively with humans determining the quality of the explainability (e.g. perhaps with mechanical turk (Poursabzi-Sangdeh et al., 2018)). We require that real-time inference be enforced and that the physics simulation be run no slower than real-time.

Finally, we can introduce artificial delays on the system actuators, storing the requested actions in an n-step queue before passing them to the simulator.

| Metric (Challenges) | Definition |
|---|---|
| Average Return | |
| Warm-Start (1) | Equation (1) |
| Time-to-$R_{min}$ (2) | Equation (2) |
| Safety Violations (4) | Equation (3) |
| Robust Performance (5) | Equation (4) |
| CVaR Return (6) | (Tamar et al., 2015b) |
| Multi-Objective (6) | Equation (5) |
| Explainability (7) | User Evals |

*Table 2.* The full set of evaluation metrics to consider to address all the challenges of real world RL, along with which challenges they are relevant for.

In addition to evaluating the average rewards achieved during the trials, we can also look at specific metrics to evaluate individual challenges (Table 2). We look at the warm-start performance after the training on the behavior policy data (Eq. 1). For sample efficiency, we examine the number of data samples required to reach a minimum performance level (Eq. 2). We also track the number of safety violations for every safety constraint (Eq. 3). We evaluate the worst-case reward over sampled perturbations (Eq. 2.5). We independently evaluate the reward on both objectives (Eq. 5). Finally, we want to qualitatively evaluate the explainability of the agent's policies. Only by looking at all these evaluators can we truly get an idea of an algorithm's aptitude for real-world use. Our goal is for this task to present a testbed for other researchers who wish to develop new algorithms that address the challenges of real world RL.

## 4. Related Work

While we covered related work specific to each challenge in the sections above, there are a few other works related to our goal of practical reinforcement learning. Hester & Stone similarly present a list of challenges for real world RL, but specifically for RL on robots. They present four challenges (sample efficiency, high-dimensional continuous state and action spaces, sensor and actuator delays, and real-time inference), all of which we include in our set of challenges as well. They do not include our other challenges such as robust performance, learning from fixed logs, non-stationarity, safety constraints, etc. Their approach is to setup a real-time architecture for model-based learning where ensembles of models are learned to improve robustness and sample efficiency.

Henderson et al. investigate the effects of existing degrees of variability between various RL setups and their effects on algorithm performance. Although restricted to the domain of existing environments, Henderson et al. propose more robust performance estimators for RL learning algorithms

which complement those presented here. The paper ends with the question "In what setting would [a given algorithm] be useful?" to which we try to contribute by proposing a specific setting in which well-adapted work should hopefully stand out. Wagstaff argue similarly to us regarding supervised ML, mentioning the too-strong lack of real-world applications in ML conferences and the subsequent impact on research directions this can have. The Horizon platform (Gauci et al., 2018) and Decision Service (Agarwal et al., 2016) provide software platforms for training, evaluation and deployment of RL agents in real-world systems. In the case of Decision Service, transition probabilities are logged to help make off-policy evaluation easier down the line, and both systems consider different approaches to off-policy evaluation. We believe well-structured frameworks such as these are crucial to productionizing RL systems. Finally, Irpan discusses a complementary series of difficulties of getting RL systems working on real systems, and offer interesting directions for future research such as taking into consideration domain-specific priors, or densifying the reward signal.

## 5. Conclusion

We argue that real-world problems in RL contain multiple challenges which are not often considered in the current literature. We have presented nine such challenges for practical RL. For each one, we have motivated the challenge, presented examples, and specified the challenge more formally. For practitioners wishing to deploy RL to their own problems, we have pointed the reader to relevant references for each challenge that may guide them in deploying RL to a production system. For RL researchers, we have presented an example environment and evaluation criteria with which to measure progress on these challenges. There have been many recent works on each of these challenges individually, but little work on approaches that address all nine challenges.

There are a few themes that appear across the related works for each challenge. While model-based RL seems especially promising to address the issue of sample efficiency, it can also help address off-policy evaluation, robustness, and explainability. Ensembles are useful both to improve sample efficiency and robustness. One very important component of deploying RL to any real product is interacting with the product owner or other human experts. We must work closely with the experts to formulate the right reward objectives and safety constraints, to get expert demonstrations to warm-start learning, and we must explain the algorithm's actions enough that they have enough confidence in the system to deploy it. These themes should point the way towards research that can address all these challenges and be deployed on more real world systems and products.

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

# Appendix

**Cart-Pole** Variables: $x, \theta$

| Type | Constraint |
|---|---|
| Static | Limit range: $x_l < x < x_r$ |
| Kinematic | Limit velocity near goal: $|\theta_c - \theta| > \theta_L \vee \dot{\theta} < \dot{\theta}_V$ |
| Dynamic | Limit cart acceleration: $\ddot{x} < A_{\max}$ |

**Walker** Variables: $\boldsymbol{\theta}, \boldsymbol{u}, \boldsymbol{F}$

| Type | Constraint |
|---|---|
| Static | Limit joint angles: $\boldsymbol{\theta}_L < \boldsymbol{\theta} < \boldsymbol{\theta}_U$ Enforce uprightness: $0 < \boldsymbol{u}_x$ |
| Kinematic | Limit joint velocities: $\max_i \left| \dot{\boldsymbol{\theta}}_i \right| < L_{\dot{\theta}}$ |
| Dynamic | Limit foot contact forces: $\boldsymbol{F}_{\text{foot}} < F_{\max}$ |

**Manipulator** Variables: $\boldsymbol{\theta}, \boldsymbol{F}, \mathcal{M}$

| Type | Constraint |
|---|---|
| Static | Limit joint angles $\boldsymbol{\theta}_L < \boldsymbol{\theta} < \boldsymbol{\theta}_U$ Avoid dynamic obstacles $\mathcal{M} \cap \mathcal{M}_{O,i} = \varnothing$ Avoid self-contact $\mathcal{M} \cap \mathcal{M} = \mathcal{M}$ |
| Kinematic | Limit joint velocities: $\max_i \left| \dot{\boldsymbol{\theta}}_i \right| < L_{\dot{\theta}}$ |
| Dynamic | Acceleration Limits: $\max_i \left| \ddot{\boldsymbol{\theta}}_i \right| < L_{\ddot{\theta}}$ Limit end effector forces: $\boldsymbol{F}_{\text{EE}} < F_{\max}$ |

**Humanoid** Variables: $\boldsymbol{\theta}, \boldsymbol{u}, \boldsymbol{F}$

| Type | Constraint |
|---|---|
| Static | Limit joint angles: $\theta_{L,i} < \boldsymbol{\theta}_i < \theta_{U,i}$ Enforce uprightness: $0 < \boldsymbol{u}_x$ |
| Kinematic | Limit joint velocities: $\max_i \left| \dot{\boldsymbol{\theta}}_i \right| < L_{\dot{\theta}}$ |
| Dynamic | Limit foot contact forces. $\boldsymbol{F}_{\text{Foot}} < F_{\max}$ Encourage falls on posterior $\boldsymbol{F}_i < F_{\max,1} \forall i \in \mathcal{C} \setminus i_{\text{post}}$ $\boldsymbol{F}_{post} < F_{\max,2}$ |

*Table 3.* Safety-constrained control environments.

## Additional Environments

We extend four example environments from the DeepMind Control Suite (Tassa et al., 2018) with both safety and non-stationarity contraints (which can be considered independently or together) to illustrate the challenges proposed in this paper. Our goal is both that these environments drive research in real-world RL as well as help evaluate candidate algorithms' applicability to real-world problems. Future work involves developing an open-source benchmark around these safety constraints.

We chose the Cart-Pole, Quadruped, Reacher, and Humanoid tasks for their increasing difficulty and closeness to real-world control systems.

## Safety

In the case of physical systems, we can generally consider that safety constraints involve one or many of static, kinematic and dynamic aspects of the system. We present safety constraints of these environments in Table 3.

## Robustness

| Env. | Noise | Non-Stationarity |
|---|---|---|
| Cart-Pole | Actuator and sensor delays | Track friction increasing with time |
| Walker | Noisy perception of terrain | Occasionally non-responsive leg actuator |
| Manipulator | Imprecise proprioception | Changes in gripper friction |
| Humanoid | Reduced torque on leg actuator | Varying payload CoGs |

*Table 4.* Possible perturbations to environments to illustrate noise & non-stationarity.

Table 4 presents some ideas of specific noise and non-stationarity perturbations for each of the proposed environments. The possibilities are effectively endless, and certain choices could make for impossible to learn, so well-designed environments will be important for useful evaluations.

## Evaluators

We resume the proposed evaluators in table 5.

| Challenge | Evaluator |
|---|---|
| Off-line | $J^{start} = R(\texttt{Train}(D_{\pi_B}))$ |
| Efficient | $J^{eff.} = \min |\mathcal{D}_i|$ s.t. $R(\texttt{Train}(D_i)) > R_{\min}$ |
| Safe | $\boldsymbol{J}^{safety}(\pi) = \left(\sum_{i=1}^{T} c_j(s_i, a_i)\right)_{1 \leq j \leq K} \in \mathbb{R}^K$ |
| Robust | $J^{robust}(\pi) = \frac{1}{K}\sum_{p \in \mathbf{P}} \mathbf{E}^p\left[\sum_{i=1}^{T} r(s_i, a_i)\right]$ |
| Discerning | $\boldsymbol{J}^{multi}(\pi) = \left(\sum_{i=1}^{T_n} r_j(s_i, a_i)\right)_{1 \leq j \leq K} \in \mathbb{R}^K$ |

*Table 5.* Proposed evaluators.