# OpenReview forum: "Challenges of Real-World Reinforcement Learning"
_ICML.cc/2019/Workshop/RL4RealLife — RL4RealLife 2019_

### Official Review · AnonReviewer1 · 2019-05-25
**A well written paper focusing on a highly relevant topic to the workshop.**

**Rating:** 5
**Confidence:** 4

**Review:**

=============
== Summary
=============

This paper presents a systematic study of the difficulties of scaling RL algorithms outside of toy/simulated domains and into the real world: it is a perfect fit for this workshop. The core of the work enumerates nine distinct challenges, each highlighting one major barrier that must be overcome in order for RL methods to be deployed in real systems. As the paper notes, research often studies various combinations of these challenges, but rarely endeavors to develop algorithms that can cope with the difficulty of all of them at once. Part of the argument of the paper suggests that more research should concentrate on this fully general problem, which is a timely and important point. A candidate environment presenting all of the challenges (and sufficient knobs to change various settings) is described as well, showcasing its potential for fruitful empirical study for those seeking to test algorithms on a difficult, real domain.

Overall, this is a well written paper focusing on a highly relevant topic to the workshop.

=============
== Questions/Comments:
=============

- Regarding the high dimensional state-action space challenge: the paper mentions "relation over the actions" as a measure for evaluating policy performance. How might action relation be measured?

- Regarding safety: it is worth citing Phil Thomas' thesis "Safe Reinforcement Learning" (2015),
(and perhaps mentioning Thomas et al. (2017)).


=============
== Misc.
=============

- It looks the Evans & Gao citation is missing a date.

=============
== References
=============

Thomas, Philip S., et al. "On ensuring that intelligent machines are well-behaved." arXiv preprint arXiv:1708.05448 (2017).

Thomas, Philip S. "Safe Reinforcement Learning". PhD Thesis, University of Massachusetts, Amherst, 2015.

---

### Official Review · AnonReviewer2 · 2019-05-25
**Good summary of real world RL challenges**

**Rating:** 5
**Confidence:** 4

**Review:**

The paper lists and explains in depth 9 difficulties in applying RL in the real world as opposed to simulated environments. The authors propose ways to measure progress on each of these aspects and a simulated environment that captures all these challenges to some degree.

I largely agree with the stated challenges, and I think the authors' view of these challenges would be better shaped by taking into account systems that are trying to do RL in the real world such as Horizon (https://arxiv.org/abs/1811.00260) and the Decision Service (https://arxiv.org/abs/1606.03966). These systems (a) log the probabilities of the behavior policy so that off policy evaluation can be unbiased (or there can be an explicit bias variance tradeoff) (b) work in epochs where the behavior policy is fixed thus sidesteping mild delays in reward and take other steps to mitigate these issues. The paper would be improved by an informed discussion of how these systems try to deal with the nine challenges.

Otherwise I think this paper is an accessible introduction for anyone aspiring to use RL in the real world.

Pros:
- Good summary of many of the challenges in real world RL
- Easy to implement metrics for tracking progress along these challenges
- An environment that captures all the challenges at once.
Cons:
- Paper does not discuss how already existing systems deal (or not) with these challenges.

---

### Decision · Program_Chairs · 2019-05-28

Accept